organometallic chemistry/materials science

butadiene, styrene, tri-block copolymer, high *cis*-1,4 unit, catalytic active centre

**Author for correspondence:**
Xiaodong Fan
e-mail: Xfand@126.com

This article has been edited by the Royal Society of Chemistry, including the commissioning, peer review process and editorial aspects up to the point of acceptance.

# A novel synthetic strategy for styrene–butadiene–styrene tri-block copolymer with high *cis*-1,4 units via changing catalytic active centres

Jie Liu[1,2], Xin Min[1], Xuan Zhang[1], Xiuzhong Zhu[1], Zichao Wang[1], Tong Wang[1] and Xiaodong Fan[1]

[1]Ministry of Education and Shaanxi Key Laboratory of Macromolecular Science and Technology, School of Science, Northwestern Polytechnical University, Xi'an, Shaanxi 710129, People's Republic of China
[2]School of Materials Science and Engineering, Shaanxi Province Key Laboratory of Catalytic Foundation and Application, Shaanxi University of Technology, Hanzhong, Shaanxi 723001, People's Republic of China

JL, 0000-0001-8499-1531; XM, 0000-0002-6701-4656

A styrene–butadiene–styrene tri-block copolymer (SBS) with a high *cis*-1,4 unit content (greater than 97%) was synthesized by a novel synthetic strategy based on changing the catalytic active centres using *n*-butyllithium and a nickel-based catalyst. Firstly, styrene was polymerized via anionic polymerization using butyllithium as the initiator (Li, activity centre Li) at 50°C. The obtained alkylated macroinitiator (PSLi) was aged with nickel naphthenate (Ni) and boron trifluoride etherate (B) to prepare a second reactive centre (Ni-F), which was used to initiate the polymerization of butadiene (Bd). Finally, triphenyl phosphine (PPh₃) was added to adjust the electron density of the third active centre (P-Ni-F), and styrene monomer was added again to synthesize the second polystyrene block to obtain SBS. The polymerization technique presented here is simple and has an efficient initiation effect due to the high initiation activities for the different monomers. It also exhibits excellent control over the stereo-structure of the butadiene segments in the prepared copolymers, and the SBS polymers with high *cis*-1,4 unit content were easily achieved.

**Scheme 1.** Synthesis routes of the SBS via a strategy based on changing catalytic active centres.

# 1. Introduction

Advanced polymer materials are widely concerned by people because of their light weight, high strength and multiple functions [1–5]. Styrene–butadiene–styrene tri-block copolymer (SBS), as one of the most important thermoplastic elastomers with excellent mechanical properties and processability, is an attractive material that has drawn significant attention in industry and academia [6–10]. Anionic polymerization is presently used in the industrial synthesis of SBS, and the obtained copolymers have narrow molecular weight distributions, but anionic polymerization cannot effectively control the stereoregularity of butadiene. Furthermore, to increase the polymerization reactivity, it is often necessary to add an electron donor such as tetrahydrofuran (THF) or tetramethylenediamine (TMEDA) to the polymerization system, which results in only a 35–40% *cis*-1,4 structure content of the polybutadiene segment [11–13].

Previous reports have shown that increasing the content of *cis*-1,4 units can significantly improve the poor low-temperature resistance and tensile properties of the polybutadiene and its block copolymers [14,15], while anionic polymerization has a severely limited ability to control the stereo-configuration of polymer segments. Matmour *et al.* [16] first reported using a dilithiated initiator in the synthesis of SBS in the absence of additives to obtain a tri-block copolymer with a high 1,4-unit content (91%), but the *cis*-1,4 structure content was only 41%.

A coordination catalyst system with a high catalytic activity and good directional function for butadiene polymerizations is an effective approach to control the stereo-structure of polybutadiene [17–19]. Due to the large polymerization activity differences between styrene with butadiene when using the same coordination catalyst, it is difficult to efficiently achieve block copolymers of these two monomers using only one catalyst. Caprio *et al.* [20] and Shiono *et al.* [21] have reported the synthesis of SBS copolymers consisting of *cis*-PB segments (*cis*-1,4 > 70%) by using metallocene-titanium and methylaluminoxane; however, the conversion ratio of the monomers was relatively low. Grubbs *et al.* [22] synthesized SBS tri-block copolymers containing 100% 1,4-microstructure PB segments by mechanistic transformation using ring-opening polymerization and atom transfer radical polymerization, and achieved a monomer conversion rate greater than 80%, but the *cis*-1,4 units only accounted for 64% of the total 1,4- structure content. Hou *et al.* [23–25] and Cui *et al.* [26] synthesized a series of styrene–butadiene/isoprene di-block or tri-block copolymers with a high percentage of *cis*-1,4 units (*cis*-1,4 > 95%) and a high conversion rate using cationic alkyl rare-earth metal catalysts. However, the complicated preparation process of these cationic complexes and the cost of the co-catalyst $[Ph_3C][B(C_6F_5)_4]$ both seriously limited their industrial applications.

Based on several literature reports detailing the synthesis of high *cis*-1,4 structure polybutadiene using rare-earth catalysts [17,19] and ternary nickel catalysts [27–29], the aim of the present study is to synthesize a SBS tri-block copolymer with a high *cis*-1,4 unit content by a novel synthetic strategy based on changing catalytic active centres. Firstly, anionic polymerization of styrene using butyllithium (Li) as the initiator was used to obtain an alkylated macromolecular initiator (PSLi). Secondly, PSLi was aged with nickel naphthenate (Ni) and boron trifluoride etherate (B), and then used to initiate the polymerization of butadiene (Bd). Finally, triphenyl phosphine (PPh₃) was added to adjust the electron density of catalytic active centre, and styrene monomer was added once again to synthesize SBS. The synthetic routes are illustrated in scheme 1. This detailed synthetic procedure based on changing catalytic active centres is a low-cost and efficient strategy to synthesize SBS tri-block copolymers containing high *cis*-1,4 unit contents, which could lead to excellent mechanical properties.

# 2. Experimental procedure

## 2.1. Materials

1,3-butadiene (Bd, 1.9 M in hexane) was obtained from Energy China and used as received. Styrene (St, Macklin, China) was purified by distillation at a reduced pressure over calcium hydride and

**Table 1.** The synthesis of SBS via changing catalytic active centres.[a]

| entry | polymer | yield[b] wt% | St cont[c] mol% | $M_n^d \times 10^4$ g mol$^{-1}$ | $M_w/M_n^d$ | cis-1,4[e] % |
|---|---|---|---|---|---|---|
| 1 | PS | 99.2 | 100 | 1.1 | 1.07 | — |
| 2 | PS-b-PB | 60.1 | 17.9 | 4.3 | 1.31 | 97.2 |
| 3 | SBS | 67.6 | 35.1 | 6.1 | 1.53 | 97.2 |

[a]1, 2, 3 correspond to the first, second, and third polymerization steps.
[b]Yield corresponds to styrene and butadiene monomer feeds.
[c]Determined by the $^1$H NMR spectrum.
[d]Measured by SEC-MALLS.
[e]Measured by $^1$H NMR and $^{13}$C NMR spectroscopy.

diluted to 2 M in cyclohexane and stored at $-18°$C until use. Nickel (II) naphthenate (Ni, 5 wt%) obtained from Meryer China and was diluted to 0.025 M in cyclohexane, and butyllithium (Li, 2.5 M in hexane) was obtained from Energy China and was used as received. Triphenyl phosphine (PPh$_3$, 97%) was obtained from Energy China and diluted to 0.5 M with cyclohexane. Boron trifluoride etherate (B, 98%) and 2,6-di-tert-butyl-p-cresol (antiager 264, CP) were purchased from Macklin China. Cyclohexane was distilled with calcium hydride and stored in a 4 Å molecular sieve for one week prior to use. Methanol (AR), ethanol (AR) and methyl ethyl ketone (AR) were received and used without further purification.

## 2.2. The synthesis of SBS

All syntheses were conducted in a dry argon atmosphere, and a detailed polymerization procedure (as shown in table 1) is described as a typical example. First, Li (0.05 mmol, 2.5 M in *n*-hexane, [Li]/[Ni] = 5) was used to initiate St (4 mmol, 2 M in cyclohexane, [St]/[Li] = 80) to perform an anionic polymerization in a Schlenk tube, with a rubber septum at 50°C for 1 h. Then, Ni solution (0.01 mmol, 0.025 M in cyclohexane) and B (0.05 mmol, [B]/[Li] = 1) were added with continuous stirring, and aged for 15 min at 50°C, and a dark brown-coloured catalyst solution was obtained. Then the catalyst solution and the butadiene solution (Bd, 6 mmol, [Bd]/[Ni] = 600) were injected into a Schlenk tube, and the polymerization was carried out at 50°C for 3 h. Finally, triphenyl phosphine (PPh$_3$, 0.01 mmol, [PPh$_3$]/[Ni] = 1) was added, and 15 min later, styrene solution (St, 2 mmol, 2 M in cyclohexane, [St]/[Ni] = 200) was added again. After 3 h, the polymerization was quenched by addition of ethanol, containing antiager 264 (1 wt%) as a stabilizer. The product was precipitated in methanol and repeatedly washed with ethanol, followed by extraction with butanone and hexane, successively. Then, the product was dried under vacuum at 40°C until a constant weight of a white solid was obtained.

## 2.3. Characterization

The macromolecular structures of the copolymers were analysed by $^1$H NMR spectroscopy (Bruker 400 MHz) in CDCl$_3$ using tetramethylsilane (TMS) as an internal standard. The actual ratio of the PS and PB segments in the copolymer was estimated by $^1$H NMR, according to a previously published method [21,26]. In addition, the ratio of 1,4- and 3,4- structure content was also determined by $^1$H NMR, and the ratio of cis-1,4 and trans-1,4 structure content was determined by $^{13}$C NMR [21,30].

FT-IR spectra were recorded using a Bruker Nicolet iS10 spectrophotometer, where the SBS films were cast onto a KBr disc.

The number-average molecular weights ($M_n$) and polydispersity indices ($M_w/M_n$) of the SBS were measured by a DAWN EOS size exclusion chromatography/multi-angle light scatter instrument (SEC-MALLS, Wyatt Technology). The polymer samples were dissolved in THF with the concentration of 5.0 mg ml$^{-1}$, and 200 µl aforementioned polymer solution was injected into the SEC instrument to conduct the measurement. HPLC grade THF was used as the eluent with a flow rate of 0.5 ml min$^{-1}$ at 25°C. The $M_w$ data obtained from this instrument are the absolute molecular weight value without a standard sample.

The glass transition temperatures ($T_g$) of copolymers were measured using differential scanning calorimetry (DSC 200 PC, Netzsch Instruments, Germany) by scanning 10 mg samples at a scan rate of 10°C min$^{-1}$ from $-130°$C to 200°C. To avoid effects of thermal hysteresis, samples were first heated to 200°C and then cooled down to $-130°$C, followed by DSC scanning in a given temperature range.

Atomic force microscopy (AFM) was used to study the surface topography of the SBS films. Images were obtained using a Dimension FastScan and Dimension Icon (Bruker) with a Si tip. The sample was prepared by adding three drops (150 µl) of a 1.0 wt% solution of the copolymer in toluene onto a Si wafer, and the sample was then analysed using AFM after drying for 48 h.

# 3. Results and discussion

## 3.1. The synthesis of SBS via Ni/PSLi/B catalyst system

Traditional Zigler–Natta type ternary nickel-based catalysts have a higher catalytic activity and *cis*-1,4 stereo-selectivity for butadiene [27,28], but a lower catalytic activity for styrene, which makes it difficult to obtain a block copolymer with a relatively high styrene content. Nevertheless, the anionic polymerization using butyllithium (Li) as the initiator has a higher catalytic activity towards styrene. Compared with sec-butyllithium and tert-butyllithium, *n*-butyllithium has better storage stability, use safety and lower price, which are benefits for industrial-scaled application of this synthetic strategy. Therefore, the *n*-butyllithium was chosen as the initiator for styrene in the polymerization step. Firstly, the macromolecular alkylation reagent PSLi containing a PS segment was synthesized by anionic polymerization, it was combined with nickel naphthenate (Ni), and boron trifluoride etherate (B) was used to initiate the polymerization of butadiene to obtain a PS-*b*-PB di-block copolymer. Since this system had almost no catalytic activity towards styrene, triphenyl phosphine (PPh₃) was added to adjust the electron density of the catalyst before the polymerization of the third stage styrene to finally obtain the SBS tri-block copolymer.

Table 1 shows that the three-step polymerization of SBS involving the changing active centres has relatively high conversion rates, and the obtained SBS has a high *cis*-1,4 unit content and a narrow molecular weight distribution ($M_w/M_n = 1.53$). The *cis*-1,4 content (97.2%) is much higher than the commercially available SBS (*cis*-1,4 = 35–40%) prepared by the commonly used lithium anion polymerization method, which is attributed to the efficient coordination capacity and stereo-selectivity of the nickel-based catalyst.

As shown in figure 1, the chemical shifts at 5.45 ppm and 5.05 ppm correspond to the 1,4- and 1,2-structures of the PB segment, whereas the chemical shifts between 6.4 and 7.2 ppm represent the benzene ring in the PS segment [21,30]. The change of the styrene and butadiene contents in the products of the three different polymerization steps shown that the homopolymer PS, the di-block copolymer PS-*b*-PB and the tri-block copolymer SBS were successfully synthesized in these three steps, respectively. In addition, the integration of the ¹H NMR peaks indicated that the SBS tri-block copolymer contains a high 1,4- unit content.

The SEC curves of PS, PS-*b*-PB and SBS products at each of the three synthetic steps are shown in figure 2. The PS-*b*-PB curve was shifted to a higher molecular weight region when the butadiene monomer was added and maintained its monomodal shape. It was shown that there was no homopolymer in the PS-*b*-PB copolymer samples. Compared to PS-*b*-PB, the SBS curve was also shifted to a higher molecular weight region upon addition of styrene monomer in the third step, which indicates that the SBS tri-block polymer was successfully prepared.

Based on the literature reports [29,31,32], the proposed mechanism of polymerization of butadiene is shown in scheme 2. Firstly, styrene was polymerized using butyllithium as an initiator via an anionic mechanism. The obtained macromolecular alkylating agent (PSLi) was aged with Ni and boron trifluoride diethylether (B) to obtain a second active centre (Ni-F). Then, this Ni-F catalyst system was used to initiate the polymerization of butadiene with high *cis*-1,4 stereo-selective, due to the coordination effect of the Ni atom. However, the catalyst system with a PS-*b*-PB segment had almost no catalytic activity towards styrene. After an equimolar amount of triphenylphosphine with Ni ($[PPh_3]/[Ni] = 1:1$) was added to adjust the electron density around the Ni atom, the third active centre (P-Ni-F) was obtained. The catalytic activity of the catalyst system towards styrene showed a significant improvement, and the conversion of styrene reached 90%. By using this synthetic strategy, an SBS tri-block copolymer with a high *cis*-1,4 structure content was finally synthesized. Table 2 presents a series of macromolecular characters of SBSs obtained via changing catalytic active centres with molar feed ratio of butadiene verse styrene monomer. By changing the amount of styrene under the butadiene fixed to 6 mmol, the effect of [St]/[Bd] feed ratio on copolymer's structure was investigated. Clearly seen from table 2, the actual percentage of the polystyrene component in copolymer could be effectively controlled via the monomer feed ratio of [St]/[Bd].

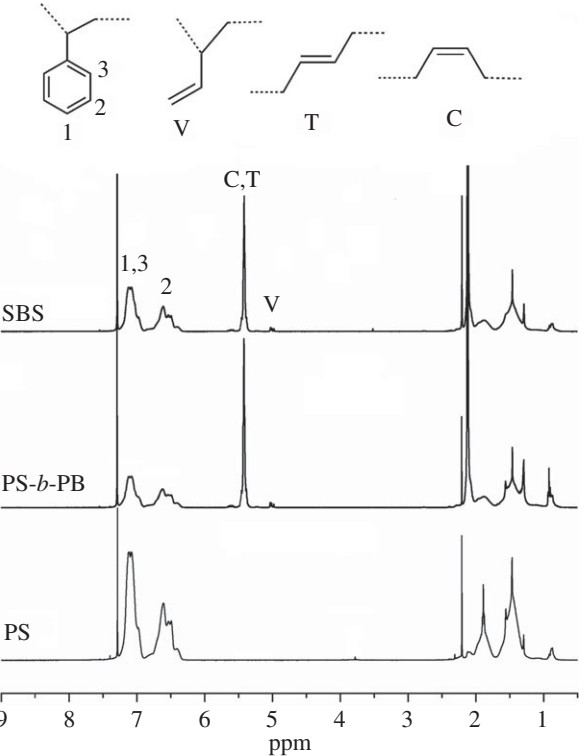

**Figure 1.** $^1$H NMR of polymers from the third synthesis steps.

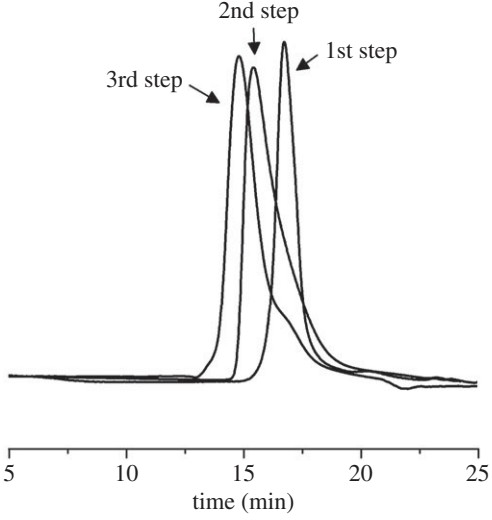

**Figure 2.** SEC trace of polymers from the three-step polymerization.

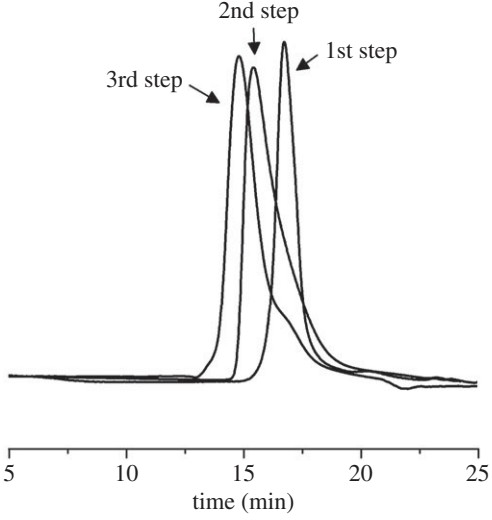

**Scheme 2.** Proposed mechanism for the synthesis of SBS via changing catalytic active centres.

**Table 2.** The effect of monomer feed ratio in the synthesis of SBS via changing catalytic active centres.[a]

| entry | $F_{St1}^b$ mmol | $F_{Bd2}^c$ mmol | $F_{St3}^d$ mmol | St cont[e] mol% | $M_n^f \times 10^4$ g mol$^{-1}$ | $M_w/M_n^f$ | cis-1,4[g] % |
|---|---|---|---|---|---|---|---|
| 1 | 2 | 6 | 1 | 21.2 | 4.6 | 1.51 | 97.2 |
| 2 | 4 | 6 | 2 | 35.1 | 6.2 | 1.53 | 97.2 |
| 3 | 6 | 6 | 3 | 43.9 | 7.9 | 1.76 | 97.3 |

[a]First step, [Li] = 0.05 mmol. Second step, [B]/[Li] = 1, [Ni] = 0.01 mmol. Third step, [PPh₃]/[Ni] = 1.
[b]The styrene content in feed of the first step.
[c]The butadiene content in feed of the second step.
[d]The styrene content in feed of the third step.
[e]Determined by the $^1$H NMR spectroscopy.
[f]Determined by SEC-MALLS.
[g]Determined by $^1$H NMR and $^{13}$C NMR spectroscopy.

**Table 3.** The effect of [Li]/[Ni] in the synthesis of SBS via changing catalytic active centres.[a]

| entry | Li/Ni | conv[b] % | St cont[c] % | $M_n^d \times 10^4$ g mol$^{-1}$ | $M_w/M_n^d$ | cis-1,4[e]% |
|---|---|---|---|---|---|---|
| 1 | 2.5 | 17.1 | 73.5 | 3.9 | 2.52 | 96.8 |
| 2 | 5 | 92.3 | 35.1 | 6.2 | 1.53 | 97.2 |
| 3 | 7.5 | 92.8 | 35.0 | 6.1 | 1.62 | 97.2 |
| 4 | 10 | 93.3 | 34.8 | 5.9 | 1.69 | 97.1 |
| 5 | 15 | 82.5 | 37.5 | 5.3 | 2.06 | 96.7 |

[a]First step, [Li] = 0.05 mmol, [St] = 4 mmol. Second step, [B]/[Li] = 1, [Ni] = 0.01 mmol, [Bd] = 6 mmol. Third step, [PPh₃]/[Ni] = 1, [St] = 2 mmol.
[b]Conversion of monomer Bd in the second step.
[c]Determined by $^1$H NMR spectroscopy.
[d]Determined by SEC-MALLS.
[e]Measured by $^1$H NMR and $^{13}$C NMR spectroscopy.

## 3.2. The effect of [Li]/[Ni]

It was found that using *n*-butyllithium versus nickel naphthenate ([Li]/[Ni]) seriously affected the conversion of butadiene during the second polymerization step of SBSs. The detailed results are shown in table 3, which shows that when the [Li]/[Ni] molar ratio was less than 2.5, the conversion of butadiene was very low, and incomplete alkylation of nickel isophthalate (Ni) possibly occurred. The experimental data indicated that a molar ratio of [Li]/[Ni] = 5 ensured a conversion of butadiene over 92%, and that the SBS would have a high cis-1,4 unit content with a narrow molecular weight distribution ($M_w/M_n$ = 1.53). In addition, the amount of alkyllithium has little effect on the stereochemistry of the PB segments in the obtained SBS, due to the coordination and orientation effects of nickel during the polymerization.

## 3.3. The effect of [PPh₃]/[Ni]

As shown in figure 3, the system displayed almost no catalytic activity towards styrene ([PPh₃]/[Ni] = 0) after completion of the butadiene polymerization. However, the electron density of catalytic active centres could be adjusted by adding a certain amount of electron-donating triphenylphosphine (PPh₃), and the polymerization activity of the system towards styrene was significantly improved. When the amount of triphenylphosphine [PPh₃]/[Ni] = 1, the catalytic activity of the catalyst system towards styrene was the highest, which allowed a conversion rate of styrene in the third polymerization step to reach values higher than 90%.

## 3.4. The characterization of SBS

The FT-IR spectrum of the SBS prepared via changing the catalytic active centres is presented in figure 4. The FT-IR spectrum exhibited two distinct absorption bands at 700 and 1500 cm$^{-1}$, which correspond to

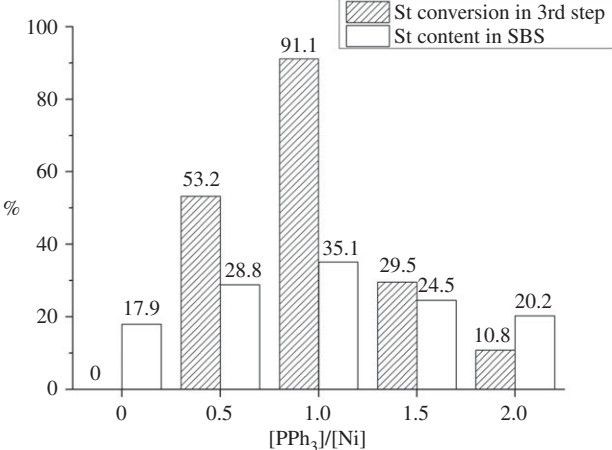

**Figure 3.** The effect of the [PPh₃]/[Ni] in the third polymerization step of SBS.

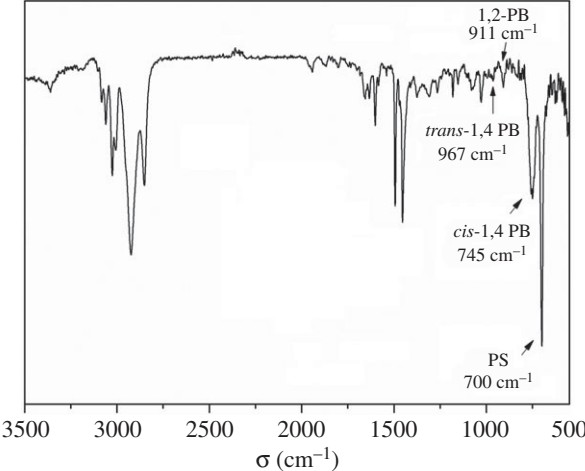

**Figure 4.** FT-IR spectrum of SBS tri-block copolymer obtained via changing catalytic active centres.

the out-of-plane deformation vibrations and skeleton stretching vibrations of the benzene ring in the SBS. The bands at 745, 967 and 911 cm$^{-1}$ represent the characteristic absorption peaks of *cis*-1,4, *trans*-1,4 and 1,2- structures of the butadiene segment, respectively. These peaks confirmed that SBS had been successfully synthesized via changing catalytic active centres. As seen in previous reports [21,30], the data revealed that the tri-block copolymer contained very high *cis*-1,4 PB unit content (*cis*-1,4 > 97%).

The $^{13}$C NMR spectrum of the SBS is presented in figure 5, and the chemical shifts at 27.5, 32.8, 34.3 and 145.3 ppm represent the *cis*-1,4, *trans*-1,4, 1,2- and PS, respectively. The integration of these peaks indicated that the PS-*b*-PB polymer contained a much higher *cis*-1,4 unit content, and the specific results are shown in tables 2 and 3.

The SBS sample obtained via changing catalytic active centres was analysed with DSC to examine its thermal properties, as shown in figure 6. The SBS obtained via changing the catalytic active centres by incorporating 35.1% PS (molar ratio, detailed in entry 3, table 1) showed a $T_g$ of $-102.1°$C, which is very close to the $T_g$ of Ni-based butadiene rubber, and was notably lower than the $T_g$ of industrial SBS obtained via the Li catalytic system (approx. $-80°$C), indicating high performance at low temperatures. The copolymer with higher *cis*-1,4 unit content inherently possesses a much lower molecular internal rotation energy and stronger intermolecular force, which could maintain a better flexibility at an extremely low temperature [18,33].

To further study the morphology and confirm the monomer distribution in the copolymers, one tri-block copolymer sample with high styrene content (entry 3, table 1, $X_{st}$ = 35.1 mol%) was analysed by AFM. The AFM micrographs in figure 7 indicate a remarkable phase-separated morphology of the hard and soft domains containing PS and *cis*-1,4-PBD, respectively. The dark phase (PS) is evenly

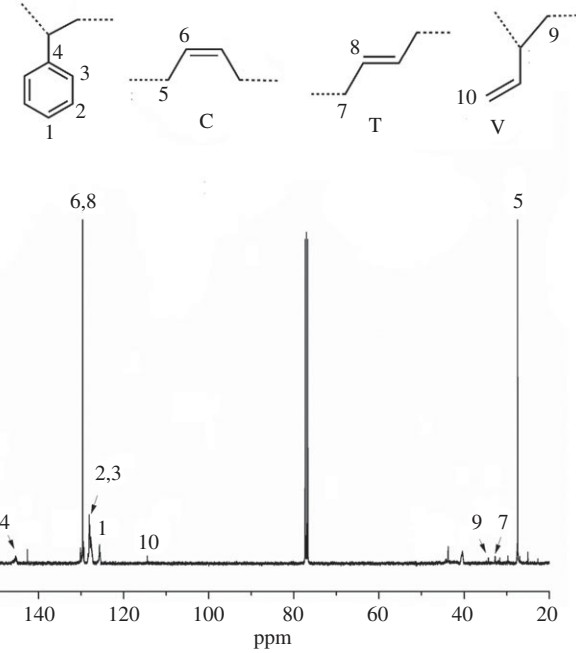

**Figure 5.** $^{13}$C NMR of SBS tri-block copolymer obtained via changing catalytic active centres.

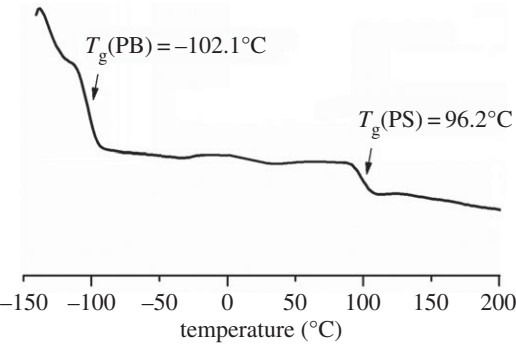

**Figure 6.** DSC curves of SBS via changing catalytic active centres (entry 3, in table 1, $X_{st} = 35.1$ mol%).

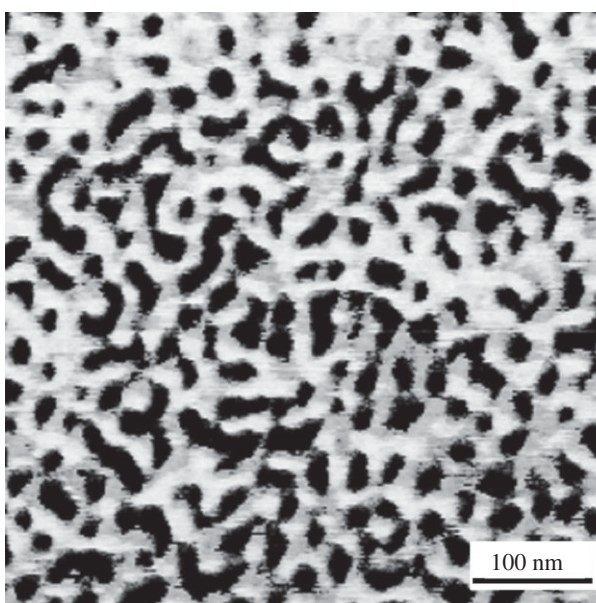

**Figure 7.** AFM micrographs of a thin film of SBS obtained via changing catalytic active centres (entry 3, in table 1, scan size: 500 nm $X_{st} = 35.1$ mol%).

distributed within the light phase (*cis*-1,4 PB), and the PS is distinctly separated from the *cis*-1,4 PB phase which further demonstrates the block architecture of the copolymer [26,34].

# 4. Conclusion

In this study, a macromolecular alkylation reagent (PSLi), obtained from the reaction between styrene and butyllithium (Li), was aged with nickel naphthenate (Ni) and boron trifluoride etherate (B) to prepare a Nickel-based catalytic system used to initiate the polymerization of butadiene. The styrene polymerization was efficiently initiated after triphenylphosphine (PPh$_3$) was added to adjust the electron density of the catalytic active centres, and a series of tri-block copolymers (SBS) with well-controlled block ratios and stereo-structures of the polybutadiene (PB) segment were successfully prepared via a novel synthetic strategy based on changing catalytic active centres. Experimental results showed that when the ratio for Li and PPh$_3$ reagents was fixed to [Li]/[Ni] = 5 and [PPh$_3$]/[Ni] = 1, the catalyst system has both a high catalytic activity towards butadiene and styrene monomers, and can also effectively control the stereo-structure of the polybutadiene segment (PB). This procedure allowed SBS tri-block copolymers with high *cis*-1,4 content (greater than 97%) to be obtained.

Data accessibility. The datasets supporting this article have been uploaded as part of the electronic supplementary material.

Authors' contributions. J.L. and X.M. contributed to the experiment design. J.L., X.Z. and T.W. performed the experiments and collected data. J.L. wrote the manuscript. J.L., X.Z., Z.W. contributed to data analysis and interpretation. X.F. provided conception and funding support. All authors gave final approval for publication.

Competing interests. We declare we have no competing interests.

Funding. This work was supported by the scientific research project of Ministry of Industry and Information Technology of P. R. China (grant no. JSJL2016140B004) and the key laboratory project of Department of Science and Technology of Shaanxi Province, China (grant no. 2013SZS17-Z02).

Acknowledgements. The authors cordially thank the Analytic and Testing Center of the Northwestern Polytechnical University for providing various characterization facilities.

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
