## [Reviewer comments · Royal Society Open Science]

Review History

RSOS-190536.R0 (Original submission)

Review form: Reviewer 1

Is the manuscript scientifically sound in its present form?

Yes

Are the interpretations and conclusions justified by the results?

Yes

Is the language acceptable?

Yes

Is it clear how to access all supporting data?

No

Do you have any ethical concerns with this paper?

No

Have you any concerns about statistical analyses in this paper?

No

Recommendation?

Major revision is needed (please make suggestions in comments)

Comments to the Author(s)

The manuscript entitled: "A novel synthetic strategy for styrene-butadiene-styrene tri-block copolymer with high cis-1,4 units via changing catalytic active centers" is a work dealing with the synthesis of SBS triblock copolymers via utilization of catalysts in order to provide a triblock copolymer elastomer with better properties. The manuscript can be accepted for publication after revision and clarification to the following points:

1. The authors state that they used n-BuLi for the synthesis of the PS block. I am wondering why they did not use sec-BuLi which is more efficient for the polymerization and leads to low molecular weight dispersities? The final polymers have high dispersities values and this can be a reason for lower homogeneity regarding the properties of the polymer.
2. Can the authors comment about the ratio of the PB different microstructures?
3. The characterization part of the SBS polymer is very short. The authors should provide more information and explanation on different points:
 - a. Why is the Tg PB significantly lower than the Li synthesized SBS? What is the reason for the lower Tg? Did the authors compare it with literature data corresponding to the same Mw?
 - b. For the AFM measurements, how was the preparation of the sample influencing the observed morphology?
 - c. also for AFM: I guess there is a mistake, "solution of the copolymer in toluene on a Si tip" it cannot be that the drop casting was done a Si tip?
 - d. GPC: how many columns and of which quality did the authors used?
 - e. GPC: on what standard is the calibration fo the SEC measurements? Was there also a RI detector used?

Review form: Reviewer 2

Is the manuscript scientifically sound in its present form?

Yes

Are the interpretations and conclusions justified by the results?

Yes

Is the language acceptable?

Yes

Is it clear how to access all supporting data?

Yes

Do you have any ethical concerns with this paper?

No

Have you any concerns about statistical analyses in this paper?

No

Recommendation?

Accept with minor revision (please list in comments)

Comments to the Author(s)

This is an interesting work about the a novel synthetic strategy for styrene-butadiene-styrene tri-block copolymer with high cis-1,4 units, which is quite interesting. However, I still have some minor concerns before I can recommend it for publication:

- 1) In Figure 7, the AFM images are not clear and do not have a clear scale bar. The authors need to have a better image with clear scale bar.
- 2) Although the work is quite interesting and could be potentially appealing to broad interest, the authors seem to have written it towards a more specialized audience. I suggest the authors add a brief overview in the introduction about the application of polymer materials, including emerging important applications, such as air filtration (ACS Appl. Mater. Interfaces, 2019, 11 (13), pp 12880–12889; Macromolecular Materials and Engineering, 2017, 302(1), 1600353-1600380; Macromolecular Materials and Engineering, 2018, 1800336-1800354) to appeal to broad interest.

Based on the above concerns, I suggest a minor revision.

Decision letter (RSOS-190536.R0)

05-Apr-2019

Dear Dr Liu:

Title: A novel synthetic strategy for styrene-butadiene-styrene tri-block copolymer with high cis-1,4 units via changing catalytic active centers

Manuscript ID: RSOS-190536

Thank you for submitting the above manuscript to Royal Society Open Science. On behalf of the Editors and the Royal Society of Chemistry, I am pleased to inform you that your manuscript will be accepted for publication in Royal Society Open Science subject to minor revision in accordance with the referee suggestions. Please find the reviewers' comments at the end of this email.

The reviewers and handling editors have recommended publication, but also suggest some minor revisions to your manuscript. Therefore, I invite you to respond to the comments and revise your manuscript.

Please also include the following statements alongside the other end statements. As we cannot publish your manuscript without these end statements included, if you feel that a given heading is not relevant to your paper, please nevertheless include the heading and explicitly state that it is not relevant to your work. We have included a screenshot example of the end statements for reference.

- Ethics statement

Please clarify whether you received ethical approval from a local ethics committee to carry out your study. If so please include details of this, including the name of the committee that gave consent in a Research Ethics section after your main text. Please also clarify whether you received informed consent for the participants to participate in the study and state this in your Research Ethics section.

OR

Please clarify whether you obtained the necessary licences and approvals from your institutional animal ethics committee before conducting your research. Please provide details of these licences and approvals in an Animal Ethics section after your main text.

OR

Please clarify whether you obtained the appropriate permissions and licences to conduct the fieldwork detailed in your study. Please provide details of these in your methods section.

- Acknowledgements

Because the schedule for publication is very tight, it is a condition of publication that you submit the revised version of your manuscript before 14-Apr-2019. Please note that the revision deadline will expire at 00.00am on this date. If you do not think you will be able to meet this date please let me know immediately.

Best wishes,
Dr Laura Smith
Publishing Editor, Journals

RSC Associate Editor:
Comments to the Author:
(There are no comments.)

RSC Subject Editor:
Comments to the Author:
(There are no comments.)

Reviewer comments to Author:
Reviewer: 1

Comments to the Author(s)

The manuscript entitled: "A novel synthetic strategy for styrene-butadiene-styrene tri-block copolymer with high cis-1,4 units via changing catalytic active centers" is a work dealing with the synthesis of SBS triblock copolymers via utilization of catalysts in order to provide a triblock copolymer elastomer with better properties. The manuscript can be accepted for publication after revision and clarification to the following points:

1. The authors state that they used n-BuLi for the synthesis of the PS block. I am wondering why they did not use sec-BuLi which is more efficient for the polymerization and leads to low molecular weight dispersities? The final polymers have high dispersities values and this can be a reason for lower homogeneity regarding the properties of the polymer.
2. Can the authors comment about the ratio of the PB different microstructures?
3. The characterization part of the SBS polymer is very short. The authors should provide more information and explanation on different points:
 - a. Why is the Tg PB significantly lower than the Li synthesized SBS? What is the reason for the lower Tg? Did the authors compare it with literature data corresponding to the same Mw?
 - b. For the AFM measurements, how was the preparation of the sample influencing the observed morphology?
 - c. also for AFM: I guess there is a mistake, "solution of the copolymer in toluene on a Si tip" it cannot be that the drop casting was done a Si tip?

- d. GPC: how many columns and of which quality did the authors used?
e. GPC: on what standard is the calibration for the SEC measurements? Was there also a RI detector used?

Reviewer: 2

Comments to the Author(s)

This is an interesting work about the a novel synthetic strategy for styrene-butadiene-styrene tri-block copolymer with high cis-1,4 units, which is quite interesting. However, I still have some minor concerns before I can recommend it for publication:

- 1) In Figure 7, the AFM images are not clear and do not have a clear scale bar. The authors need to have a better image with clear scale bar.
- 2) Although the work is quite interesting and could be potentially appealing to broad interest, the authors seem to have written it towards a more specialized audience. I suggest the authors add a brief overview in the introduction about the application of polymer materials, including emerging important applications, such as air filtration (ACS Appl. Mater. Interfaces, 2019, 11 (13), pp 12880–12889; Macromolecular Materials and Engineering, 2017, 302(1), 1600353-1600380; Macromolecular Materials and Engineering, 2018, 1800336-1800354) to appeal to broad interest.

Based on the above concerns, I suggest a minor revision.

Author's Response to Decision Letter for (RSOS-190536.R0)

See Appendix A.

RSOS-190536.R1 (Revision)

Review form: Reviewer 1

Is the manuscript scientifically sound in its present form?

Yes

Are the interpretations and conclusions justified by the results?

Yes

Is the language acceptable?

Yes

Is it clear how to access all supporting data?

Yes

Do you have any ethical concerns with this paper?

No

Have you any concerns about statistical analyses in this paper?

No

Recommendation?

Accept with minor revision (please list in comments)

Comments to the Author(s)

The answers of the authors to the review questions are quite nice and they are clarifying almost all the points. Thus the authors did not elaborate quite a lot of the answers into their manuscript although the discussion is very helpful. Additionally, regarding GPC the authors should state that no standard is used for the presented Mw data. Also, why was the sample filtered before GPC? This is something that can alter the results in case of high molecular weight dispersities. Did the authors check the quality of the solution before running the GPC test? Was the solubility fine?

Review form: Reviewer 2

Is the manuscript scientifically sound in its present form?

Yes

Are the interpretations and conclusions justified by the results?

Yes

Is the language acceptable?

Yes

Is it clear how to access all supporting data?

Yes

Do you have any ethical concerns with this paper?

No

Have you any concerns about statistical analyses in this paper?

No

Recommendation?

Accept as is

Comments to the Author(s)

The authors have well addressed my concerns and I suggest accept as it is.

Decision letter (RSOS-190536.R1)

24-Apr-2019

Dear Dr liu:

Title: A novel synthetic strategy for styrene-butadiene-styrene tri-block copolymer with high cis-1,4 units via changing catalytic active centers
Manuscript ID: RSOS-190536.R1

Thank you for submitting the above manuscript to Royal Society Open Science. On behalf of the Editors and the Royal Society of Chemistry, I am pleased to inform you that your manuscript will be accepted for publication in Royal Society Open Science subject to minor revision in accordance with the referee suggestions. Please find the reviewers' comments at the end of this email.

The reviewers and handling editors have recommended publication, but also suggest some minor revisions to your manuscript. Therefore, I invite you to respond to the comments and revise your manuscript.

Because the schedule for publication is very tight, it is a condition of publication that you submit the revised version of your manuscript before 03-May-2019. Please note that the revision deadline will expire at 00.00am on this date. If you do not think you will be able to meet this date please let me know immediately.

Once again, thank you for submitting your manuscript to Royal Society Open Science. The chemistry content of Royal Society Open Science is published in collaboration with the Royal

Society of Chemistry. I look forward to receiving your revision. If you have any questions at all, please do not hesitate to get in touch.

Best wishes,

Dr Laura Smith
Publishing Editor, Journals

RSC Associate Editor:
Comments to the Author:
(There are no comments.)

RSC Subject Editor:
Comments to the Author:
(There are no comments.)

Reviewer comments to Author:
Reviewer: 2

Comments to the Author(s)
The authors have well addressed my concerns and I suggest accept as it is.

Reviewer: 1

Comments to the Author(s)
The answers of the authors to the review questions are quite nice and they are clarifying almost all the points. Thus the authors did not elaborate quite a lot of the answers into their manuscript although the discussion is very helpful. Additionally, regarding GPC the authors should state that no standard is used for the presented Mw data. Also, why was the sample filtered before GPC? This is something that can alter the results in case of high molecular weight dispersities. Did the authors check the quality of the solution before running the GPC test? Was the solubility fine?

Author's Response to Decision Letter for (RSOS-190536.R1)

See Appendix B.

RSOS-190536.R2 (Revision)

Review form: Reviewer 1

Is the manuscript scientifically sound in its present form?

Yes

Are the interpretations and conclusions justified by the results?

Yes

Is the language acceptable?

Yes

Is it clear how to access all supporting data?

Yes

Do you have any ethical concerns with this paper?

No

Have you any concerns about statistical analyses in this paper?

No

Recommendation?

Accept as is

Comments to the Author(s)

The revised manuscript fulfills now the requirement for acceptance for publishing.

Decision letter (RSOS-190536.R2)

22-May-2019

Dear Dr liu:

Title: A novel synthetic strategy for styrene-butadiene-styrene tri-block copolymer with high cis-1,4 units via changing catalytic active centers

Manuscript ID: RSOS-190536.R2

It is a pleasure to accept your manuscript in its current form for publication in Royal Society Open Science. The chemistry content of Royal Society Open Science is published in collaboration with the Royal Society of Chemistry.

RSC Associate Editor:
Comments to the Author:
(There are no comments.)

RSC Subject Editor:
Comments to the Author:
(There are no comments.)

Reviewer(s)' Comments to Author:
Reviewer: 1

Comments to the Author(s)
The revised manuscript fulfills now the requirement for acceptance for publishing.

Appendix A

Response to Reviewers

Dear editor,

We sincerely appreciate you and the reviewers for reviewing our manuscript (Manuscript ID: RSOS-190536) entitled "A novel synthetic strategy for styrene-butadiene-styrene tri-block copolymer with high *cis*-1,4 units via changing catalytic active centers". According to the reviewers' comments and your suggestions, the manuscript has been revised carefully and substantially. The point-by-point responses to reviewers' are described as follows and marked with red font in the revised manuscript.

Reviewer: 1

Q1: *The authors state that they used n-BuLi for the synthesis of the PS block. I am wondering why they did not use sec-BuLi which is more efficient for the polymerization and leads to low molecular weight dispersities? The final polymers have high dispersities values and this can be a reason for lower homogeneity regarding the properties of the polymer.*

Response:

Thank you for your suggestion, we agree with your viewpoint. Sec-BuLi is more efficient for the polymerization than n-BuLi, and it does have the potential to obtain block polymers with lower molecular weight dispersities. However, the sec-BuLi is more active chemically, the storage and polymerization conditions are more demanding, and the price of sec-BuLi is much expensive, so n-BuLi was chose as the initiator for styrene in the first polymerization step in our experiment. In addition, in the second polymerization step in this experiment, the polymerization of butadiene was initiated by the catalytic system centered on transition metal nickel. Generally, such a similar ternary nickel-based coordination catalytic system tends to have a broader molecular weight distribution than anionic polymerization, which is also one of the reasons why the final polymers have high dispersities values in this experiment.

Q2: Can the authors comment about the ratio of the PB different microstructures?

Response:

This suggestion is very useful to us. Usually, there are three different microstructures exist in polybutadiene segment backbone, which are *cis*-1,4 unit, *trans*-1,4 unit and 1,2- unit as shown in Scheme 1, thereinto, *cis*-1,4 unit and *trans*-1,4 unit are collectively called 1,4- unit. As a consequence of differing microstructure, the various polybutadiene (PB) exhibit different physical and mechanical properties. Specifically, the high *cis*-1,4 polybutadiene exhibit good low-temperature properties, high resilience over a broad temperature range, low heat-build-up on repeated deformation and high abrasion resistance, which are highly desirable properties for tire; 1,2- PBd rubber possesses low rolling resistance and wet skid resistance; *trans*-1,4 PBd is prone to be crystallized at room temperature, which exhibits resinous status and plastic properties. (J. Appl. Polym. Sci. 2014, 131(8), pp 40153, doi: 10.1002/APP.40153; Adv. Polym. Sci. 2006, 204, pp 1–12, doi: 10.1007/12_094; Angew. Chem. Int. Ed. 119, pp 1941-1945, doi: 10.1002/ange.200604348)

Scheme 1 Three different microstructure of polybutadiene

Q3: The characterization part of the SBS polymer is very short. The authors should provide more information and explanation on different points.

a. Why is the T_g PB significantly lower than the Li synthesized SBS? What is the reason for the lower T_g ? Did the authors compare it with literature data corresponding to the same M_w ?

Response:

Thank you very much for your suggestion. In Q2, we have introduced that the microstructure of PB include *cis*-1,4 unit, *trans*-1,4 unit and 1,2- unit. Literature reported that the T_g of *cis*-1,4 PBd ,*trans*-1,4 PBd and 1,2-PBd are -106°C , -107°C and -15°C respectively. (Journal of Polymer Science, 40(136), pp 121-131, doi: 10.1002/pol.1959.1204013609) Consequently, the T_g of the SBS obtained via changing catalytic active centers in this experiment was significantly lower than the *Li synthesized SBS*, because the 1,2- unit of SBS synthesized via n-butyllithium system could account to as high as 10%, which was Significantly higher than the SBS in this experiment(1,2- structure content $\sim 2\%$).

The T_g of a polymer is determined by the influence of intramolecular and intermolecular. Because copolymer with higher *cis*-1,4 unit content or tran-1,4 which inherently possesses a much lower molecular internal rotation energy can naturally exhibit a much lower T_g which could maintain a nice flexibility at an extremely low temperature. Thereinto, *trans*-1,4 PBd was prone to crystallize, we expect to obtain a PBd with a high *cis*-1,4 structure to obtain a polymer with better mechanical properties at low temperature. Meanwhile, with the increase of the 1,2- structure content of polybutadiene, the force in the polymer molecule enhances, and the intermolecular force weakens, consequently, the T_g rised with the increase of the 1,2- structure content.(Adv. Polym. Sci. 2006, 204, pp 1–12, doi: 10.1007/12_094; *Macromol. Chem. Phys.*, 1800479, 1, doi:org/10.1002/macp.201800479)

According to the suggestion of reviewer, in order to illustrate the effect of stereo-structure on the T_g of the polymers containing PB segment, we have supplemented the sentence “Because the copolymer with higher *cis*-1,4 unit content which inherently possesses a much lower molecular internal rotation energy and stronger intermolecular force, which could maintain a better flexibility at an extremely low temperature” with red font in section 3.4 in the revised manuscript.

According to the literatures, Zhang et al. (Polymer 2009, 50, pp 5427-5433, doi:10.1016/j.polymer.2009.09.070) synthesized a SBS with 8% 1,2- structure content by a di-lithium catalyst via anionic polymerization, the M_n the of the result polymer

was 9.84×10^4 , the T_g was about $-88 \text{ }^\circ\text{C}$. Lee et al. (*Macromolecules* 2013, 46, pp 4529-4539, doi:10.1021/ma400479b.) synthesized an alternating Poly(styrene-*b*-butadiene) multiblock copolymers using a combination of living anionic polymerization and urethane based polycondensation, the M_n of the result polymer was 9.0×10^4 , the T_g of PBd segment was about $-80 \text{ }^\circ\text{C}$. While Cui et al. synthesized a SBS with M_n of 8.8×10^4 and 95.5% *cis*-1,4 structure, the T_g of PBd segment was $-104 \text{ }^\circ\text{C}$ (*Chem-Eur. J.*, 16, 14007-14015, doi:10.1002/chem.201001634), The T_g was very close to the tri-block copolymer obtained in our experiment ($M_n=6.1 \times 10^4$, *cis*-1,4:97.2%, $T_g(\text{PB})=-102.1 \text{ }^\circ\text{C}$). It is indicated that the SBS could be obtained by the coordination polymerization with lower T_g than that obtained via the lithium-based anionic coordination.

b. For the AFM measurements, how was the preparation of the sample influencing the observed morphology?

Response:

During the preparation process of the sample for AFM measurements, we found that the observed morphology was affected by the following factors, such as the solvent kinds, the concentration of copolymer solution, the coating thickness of the sample, and so on. Thereinto, the factors of the solvent kinds and the concentration of copolymer solution could affect the dispersion status of the polymer in the solvent and evaporation rate, which will further affect morphology of the coating. In addition, a suitable coating thickness is also very important in this test, and it directly determines whether a clear and usable morphology image could be obtained.

c. also for AFM: I guess there is a mistake, "solution of the copolymer in toluene on a Si tip" it cannot be that the drop casting was done a Si tip?

Response:

Thank you so much for your suggestion, and we really apologize for the inconvenience to you caused by these inaccurate expressions. The sentence of "Samples were prepared by adding three drops (150 μL) of a 1.0 wt % solution of the

copolymer in toluene on a Si *tip*" had been corrected to "Sample was prepared by adding three drops (150 μ L) of a 1.0 wt % solution of the copolymer in toluene onto a Si *wafer*" by "Track Changes" function. (as shown in section 2.3 in the revised manuscript.)

d. GPC: how many columns and of which quality did the authors used?

Response:

In the test of the GPC, 5 mg SBS was dissolved in tetrahydrofuran (THF) to obtain a solution with concentration of ca. 5 wt % and then filtered through a 0.22 mm filter. Thereinto, 200 μ l aforementioned SBS solution was injected into the SEC measurements to conduct the measurement of M_n and M_w/M_n of the tri-block polymer.

e. GPC: on what standard is the calibration for the SEC measurements? Was there also a RI detector used?

Response:

The molecular weight and polydispersity index of SBSs were determined on a DAWN EOS size exclusion chromatography/multi-angle laser light scattering instrument (SEC-MALLS, Wyatt Technology). The chromatographic system consisted of a Waters 515 pump, a differential refractometer (Optilab rEX), and a column (MZ 103 Å 300 \times 8.0 mm). The M_n data obtained from this instrument are the absolute molecular weight value without a standard sample. In addition, there also a RI detector used in the instrument, and the SEC curve in our manuscript was obtained based on the data from the RI detector.

Reviewer: 2

Q1: *In Figure 7, the AFM images are not clear and do not have a clear scale bar. The authors need to have a better image with clear scale bar.*

Response:

Thank you very much for your suggestion, and we really apologize for any

inconvenience to you because of the inappropriate image. We have processed the image according to the reviewer's suggestion, as shown in Fig.1 (Fig.7 in the revised manuscript)

Fig. 1. AFM micrographs of a thin film of SBS obtained via changing catalytic active centers. (entry 3, in table 1, scan size: 500 nm x_{Si} = 35.1 mol%)

Q2: *Although the work is quite interesting and could be potentially appealing to broad interest, the authors seem to have written it towards a more specialized audience. I suggest the authors add a brief overview in the introduction about the application of polymer materials, including emerging important applications, such as air filtration (ACS Appl. Mater. Interfaces, 2019, 11 (13), pp 12880–12889; Macromolecular Materials and Engineering, 2017, 302(1), 1600353-1600380; Macromolecular Materials and Engineering, 2018, 1800336-1800354) to appeal to broad interest.*

Response:

Thank you very much for your suggestion. According to the reviewer's suggestion, in order to further illustrate the advantage of the advanced polymer materials, we had supplemented the sentence at the beginning of introduction as follows "Advanced polymer materials are widely concerned by people because of their lightweight, high strength and multiple functions" with red font in the revised

manuscript.

In addition, the sections of “Ethics” and “Acknowledgements” had been supplemented at the end of the revised manuscript according to the suggestion of the editor.

Finally, this article has been carefully checked repeatedly, and the grammatical and wording errors had been corrected and marked using "Track Changes" function in the revised manuscript.

We hope that the explanations and modifications fulfill the reviewers' requirements and the manuscript meets the criteria to be accepted for publication in *Royal society Open Science*.

If there are any problems, please do not hesitate to inform us as soon as possible. Finally, we take this opportunity to say thanks to you and the reviewers.

Yours sincerely

Xiaodong Fan, et al.

Appendix B

Response to Reviewers

Dear editor,

We sincerely appreciate you and the reviewers for reviewing our manuscript (Manuscript ID: RSOS-190536.R1) entitled "A novel synthetic strategy for styrene-butadiene-styrene tri-block copolymer with high *cis*-1,4 units via changing catalytic active centers". According to the reviewers' comments and your suggestions, the manuscript has been revised carefully and substantially. The point-by-point responses to reviewers' are described as follows and marked with red font and yellow highlight in the revised manuscript.

Reviewer: 1

Q1: *The answers of the authors to the review questions are quite nice and they are clarifying almost all the points. Thus the authors did not elaborate quite a lot of the answers into their manuscript although the discussion is very helpful.*

Response:

Thank you for your suggestion. We agree with your viewpoint, and the following modifications were made.

According to the Q1 in the first version of reviewer comment "*The authors state that they used n-BuLi for the synthesis of the PS block. I am wondering why they did not use sec-BuLi which is more efficient for the polymerization and leads to low molecular weight dispersities?* The final polymers have high dispersities values and this can be a reason for lower homogeneity regarding the properties of the polymer", we have supplemented the sentence "Compared with sec-butyllithium and tert-butyllithium, n-butyllithium has better storage stability, using safety and lower price, which are benefit for industrial-scaled application of this synthetic strategy. Therefore, the n-butyllithium was chosen as the initiator for styrene in the polymerization step" with red font and yellow highlight. (In the section of 3.1 in the revised manuscript 2)

According to the Q2 in the first version of reviewer comment “*can the authors comment about the ratio of the PB different microstructures*”, we briefly introduced the advantages of the cis-1,4 structure in the “introduction” section of the original manuscript, such as “Previous reports have shown that increasing the content of cis-1,4 units can significantly improve the poor low temperature resistance and tensile properties of the polybutadiene and its block copolymers”, and so on. In addition, we have already supplemented the sentence in this revised manuscript 1 “ Because the copolymer with higher cis-1,4 unit content which inherently possesses a much lower molecular internal rotation energy and stronger intermolecular force, which could maintain a better flexibility at an extremely low temperature” with red font in section 3.4 of the revised manuscript 1.

According to the Q3a in the first version of reviewer comment “ *Why is the T_g PB significantly lower than the Li synthesized SBS? What is the reason for the lower T_g ? Did the authors compare it with literature data corresponding to the same M_w ?*”, we we have provided the elaboration “The SBS obtained via changing the catalytic active centers by incorporating 35.1% PS (molar ratio, detailed in Entry 3, Table 1) showed a T_g of -102.1 °C, which is very close to the T_g of Ni-based butadiene rubber, and was notably lower than the T_g of industrial SBS obtained via the Li catalytic system (~ -80 °C), indicating high performance at low temperatures. Because the copolymer with higher cis-1,4 unit content which inherently possesses a much lower molecular internal rotation energy and stronger intermolecular force, which could maintain a better flexibility at an extremely low temperature ” in the section 3.4 of the revised manuscript 1.

Accroding to the Q3d in the first version of reviewer comment “ *how many columns and of which quality did the authors used?* ” , we have supplement the sentence “The samples were dissolved in THF with the concentration of 5.0 mg mL⁻¹. Thereinto, 200 µL aforementioned SBS solution was injected into the SEC

measurements to conduct the measurement.” in the section 2.3 of revised manuscript 2 with red font and yellow highlight.

Q2: *Additionally, regarding GPC the authors should state that no standard is used for the presented M_w data.*

Response:

This suggestion is very useful to us. We have supplemented the sentence “The M_w data obtained from this instrument are the absolute molecular weight value without a standard sample.” In the in the section 2.3 of revised manuscript 2 with red font and yellow highlight.

Q3: *Also, why was the sample filtered before GPC? This is something that can alter the results in case of high molecular weight dispersities. Did the authors check the quality of the solution before running the GPC test? Was the solubility fine?*

Response:

Thank you very much for your suggestion. Firstly, the purpose of filtration is only to remove the impurities mixed in the preparation of the polymer solution sample, such as dust in the air, and so on. Because this Sec-Malls used in this test is a precision instrument, especially the chromatography column part (MZ 103 Å 300× 8.0 mm). In order to protect the instrument, all the samples need to be filtered before being tested, otherwise, some impurities or dust may block the gel column. However, the filtration would not affect the polymer test results.

We have tested the solubility of SBS in the solution of tetrahydrofuran (THF), and the result showed that the SBS tri-block copolymers synthesized in our experiment possessed very good solubility in THF. In addition, the polymer in THF solution prepared in this GPC testing has a concentration of only 5 mg mL⁻¹, which was very low. The low concentration ensured that the polymer chain can be

sufficiently dispersed in the solvent in the random and unfold status, and the intermolecular forces was very small. Because the filter size is large enough (0.22 μm), the polymer molecular chains could completely pass the filter under a certain pressure by extrusion.

In order to verify the changes of solution state before and after filtration, a particle size test was conducted. The hydrodynamic diameter, diameter distribution were determined by Dynamic light scattering (DLS, Malvern Instruments, UK). The scattered light of a He-Ne laser at 633 nm was measured at an angle of 173° . The sample of SBS (Entry 3, in Table 1 of revised manuscript 2) was dissolved in THF with the concentration of 5.0 mg mL^{-1} . Firstly, the sample was tested directly without filtration, and then the solution was filtered through a 0.22 μm filter, and which was tested again.

Fig.1 DLS traces comparison of SBS solution in THF before and after filtration.

As shown in Fig.1, the SBS solution before and after filtration had almost no other particle size except for a certain amount of particles with a particle size of about 10 nm, and the state change before and after filtration was very small, which indicated that the polymer SBS has a good solubility in solution and was distributed

in a unfold status. In addition, the particles with a particle size of about 10 nm could easily pass through the 0.22 μm filter.

Finally, this article has been carefully checked repeatedly, and the grammatical and wording errors had been corrected and marked using "Track Changes" function in the revised manuscript.

We hope that the explanations and modifications fulfill the reviewers' requirements and the manuscript meets the criteria to be accepted for publication in *Royal society Open Science*.

If there are any problems, please do not hesitate to inform us as soon as possible. Finally, we take this opportunity to say thanks to you and the reviewers.

Yours sincerely

Xiaodong Fan, et al.